## [Peer Review File · The EMBO Journal]

Direct receptor competition gates RGL2 proteolysis for seed germination timing in Arabidopsis

Kaili Nie, Juntao Jiang, Changgen Xie, Hongyun Zhao, and Yuan Zheng

Corresponding author(s): Yuan Zheng (zhengyuan@nwafu.edu.cn)

Review Timeline:

Submission Date:	5th Jul 25
Editorial Decision:	15th Aug 25
Revision Received:	27th Oct 25
Editorial Decision:	4th Dec 25
Revision Received:	17th Dec 25
Accepted:	8th Jan 26

Editor: William Teale

Transaction Report:

Dear Prof. Zheng,

Thank you again for the submission of your manuscript entitled "Direct receptor competition gates RGL2 proteolysis for seed germination timing" and for your patience during the review process. We have now received the reports from the referees, which I copy below.

Based on the overall interest they express, I would like to invite you to address the comments of all referees in a revised version of the manuscript. I should add that it is The EMBO Journal policy to allow only a single major round of revision and that it is therefore important to resolve the main concerns at this stage. I believe the concerns of the referees are reasonable and addressable, but please contact me if you have any questions, need further input on the referee comments or if you anticipate any problems in addressing any of their points; I can offer a Zoom chat to discuss your revisions at any time. Please, follow the instructions below when preparing your manuscript for resubmission.

I would also like to point out that as a matter of policy, competing manuscripts published during this period will not be taken into consideration in our assessment of the novelty presented by your study ("scooping" protection). We have extended this 'scooping protection policy' beyond the usual 3 month revision timeline to cover the period required for a full revision to address the essential experimental issues. Please contact me if you see a paper with related content published elsewhere to discuss the appropriate course of action.

Again, please contact me at any time during revision if you need any help or have further questions.

Thank you very much again for the opportunity to consider your work for publication. I look forward to your revision.

Best regards,

William

William Teale, Ph.D.
Editor
The EMBO Journal

When submitting your revised manuscript, please carefully review the instructions below and include the following items:

2) individual production quality figure files as .eps, .tif, .jpg (one file per figure).

3) a .docx formatted letter INCLUDING the reviewers' reports and your detailed point-by-point response to their comments. As part of the EMBO Press transparent editorial process, the point-by-point response is part of the Review Process File (RPF), which will be published alongside your paper.

4) a complete author checklist, which you can download from our author guidelines ([https://wol-prod-cdn.literatumonline.com/pb-assets/embo-site/Author Checklist%20-%20EMBO%20J-1561436015657.xlsx](https://wol-prod-cdn.literatumonline.com/pb-assets/embo-site/Author%20Checklist%20-%20EMBO%20J-1561436015657.xlsx)). Please insert information in the checklist that is also reflected in the manuscript. The completed author checklist will also be part of the RPF.

6) We require a 'Data Availability' section after the Materials and Methods. Before submitting your revision, primary datasets produced in this study need to be deposited in an appropriate public database, and the accession numbers and database listed under 'Data Availability'. Please remember to provide a reviewer password if the datasets are not yet public (see <https://www.embopress.org/page/journal/14602075/authorguide#datadeposition>). If no data deposition in external databases is needed for this paper, please then state in this section: This study includes no data deposited in external repositories. Note that the Data Availability Section is restricted to new primary data that are part of this study.

Note - All links should resolve to a page where the data can be accessed.

8) For data quantification: please specify the name of the statistical test used to generate error bars and P values, the number (n) of independent experiments (specify technical or biological replicates) underlying each data point and the test used to calculate p-values in each figure legend. The figure legends should contain a basic description of n, P and the test applied. Graphs must include a description of the bars and the error bars (s.d., s.e.m.).

9) We would also encourage you to include the source data for figure panels that show essential data. Numerical data can be provided as individual .xls or .csv files (including a tab describing the data). For 'blots' or microscopy, uncropped images should be submitted (using a zip archive or a single pdf per main figure if multiple images need to be supplied for one panel). Additional information on source data and instruction on how to label the files are available at .

10) We replaced Supplementary Information with Expanded View (EV) Figures and Tables that are collapsible/expandable online (see examples in <https://www.embopress.org/doi/10.15252/embj.201695874>). A maximum of 5 EV Figures can be typeset. EV Figures should be cited as 'Figure EV1, Figure EV2' etc. in the text and their respective legends should be included in the main text after the legends of regular figures.

12) Our journal encourages inclusion of *data citations in the reference list* to directly cite datasets that were re-used and obtained from public databases. Data citations in the article text are distinct from normal bibliographical citations and should directly link to the database records from which the data can be accessed. In the main text, data citations are formatted as follows: "Data ref: Smith et al, 2001" or "Data ref: NCBI Sequence Read Archive PRJNA342805, 2017". In the Reference list, data citations must be labeled with "[DATASET]". A data reference must provide the database name, accession number/identifiers and a resolvable link to the landing page from which the data can be accessed at the end of the reference. Further instructions are available at .

13) In order to increase the reproducibility and reach of your work, The EMBO Journal includes a table of reagents that were used in the study. Please provide this along with your revisions.

We realize that it is difficult to revise to a specific deadline. In the interest of protecting the conceptual advance provided by the work, we recommend a revision within 3 months (13th Nov 2025). Please discuss the revision progress ahead of this time with the editor if you require more time to complete the revisions.

Referee #1:

This manuscript elucidates a receptor competition paradigm in which RGL2 functions as a central hub for competitive receptor occupancy by the GA receptor GID1 and ABA receptors PYLs. This competition directly gates RGL2 degradation through the CUL4-DDB1 substrate receptor DWA1. The discoveries of hormone-tuned receptor competition and DWA1 as a receptor-regulated degradation conduit significantly advance our understanding of GA/ABA crosstalk beyond classical antagonism. Major Comments:

1. While PYLs-DWA1 direct physical interaction is demonstrated (Fig. 5), however, the molecular mechanism of the PYLs sequestration of DWA1 remains undefined. Do PYLs, RGL2, and DWA1 form a heterotrimeric complex? Do PYLs and RGL2 competitively bind to DWA1?

Suggestion: Perform competitive LCI and Co-IP assays to determine whether PYLs and RGL2 competitively interact with DWA1, and if PYL4 overexpression rescues accelerated RGL2 degradation in 35S:Flag-DWA1 plants (linking sequestration to phenotype).

2. All 14 PYLs can interact with RGL2 and DWA1 (Fig. 1A, 5A), moreover, ABA treatment enhanced binding intensity, particularly for PYL11 (Fig. 1C), but functional data (stabilization, competition) use only PYL4. The functional conservation among PYLs is unverified.

Suggestion: Include at least one extra PYL (e.g. PYL11) in degradation (Fig. 1D) and competition assays (Fig. 2C-F).

3. Quantitative interaction assays (LCI/BiFC) must be accompanied by expression validation of all fusion partners. Consequently, the experimental results regarding changes in interaction strength are scientifically invalid without expression controls (e.g. Figure 1C, D and Figure 2A, C, D, etc.).

Suggestion: Supplementary western blot results corresponding to the specific experiments must be provided, such as the nLUC-PYL and cLUC-RGL2 protein expression levels in Figure 1 C.

4. In Figure 4, there is a lack of genetic evidence demonstrating that DWA1 executes GA-dependent RGL2 ubiquitination.

Suggestion: Generate hybrid materials by crossing 35S:Flag:DWA1 transgenic lines with GA biosynthesis mutants to provide data on protein degradation rate and ubiquitination status of RGL2 in GA biosynthesis mutants.

5. In Figure 4E, I notice the absence of detectable change of RGL2 ubiquitination levels in both WT and 35S:Flag:DWA1 seedlings without exogenous GA treatment, how does this occur in the presence of endogenous GA?

Suggestion: WT and 35S:Flag:DWA1 seedlings could undergo MG132+PAC{plus minus}GA co-treatments to eliminate confounding effects of endogenous GA on DWA1-mediated RGL2 ubiquitination levels.

6. In Figure 6B, I find substantial variation in the initial MBP-RGL2 protein levels among seedling protein extracts, Furthermore, all MBP-RGL2 Western blot results should be displayed on a single membrane to reliably demonstrate degradation rate differences.

Suggestion: Load the protein samples containing MBP-RGL2 fusion protein on a single membrane for comparative analysis while adding more time points (0, 20, 40, 60 min).

7. The manuscript aims to elucidate the competitive interplay between RGL2 and GID1 versus PYL, revealing a mechanism

distinct from the classical GA/ABA antagonism in seed germination regulation. While PYLs directly stabilize RGL2 is demonstrated (Figure 1), how GID1 directly destabilize RGL2 remains unclear (mediated by GID1-DWA module, distinct from the canonical GID1-SLY1 pathway).

Suggestion: Perform assays to demonstrate GID1 promotion of DWA1-mediated ubiquitination (like Figure 6A) and compare MBP-RGL2 degradation rates among extracts from WT, Flag-DWA1, *gid1a gid1b gid1c* and *gid1a gid1b gid1c* Flag-DWA1 seeding (like Figure 6).

8. Following the previous question, how to distinguish DWA1-mediated RGL2 degradation with canonical GA-GID1-SLY1 pathways?

Suggestion: Perform assays to demonstrate if GA-DWA1-mediated RGL2 degradation exists in *sly1* mutant.

Minor Comments:

1. The manuscript contains several labeling inaccuracies that require correction. Please verify whether nLUC-PYLs+cLUC-DWA1+ABA and nLUC-PYLs+cLUC-DWA1+GA should be nLUC-PYLs+cLUC-RGL2+ABA and nLUC-PYLs+cLUC-RGL2+GA due to a labeling error in Figure 1 C.
2. To compare the differences in MYC-RGL2 protein stability in different seedlings in Figure 1D and Figure 4C, it is necessary to provide qPCR data demonstrating consistent transcriptional levels of MYC-RGL2.
3. In Figure 2B, to demonstrate the difference in binding capacity between RGL2 and GID1 versus PYL, a more robust approach is to use either MST (MicroScale Thermophoresis) or ITC (Isothermal Titration Calorimetry) experiments to quantify the differences in their affinity constants.
4. *pyr1 pyl124* quadruple mutants should be *pyr1 pyl1 pyl2 pyl4* (list all genes) according to Park et al., 2009.

Referee #2:

Nie et al report in this manuscript a competitive mechanism for RGL2 stabilization/ proteasomal degradation by the ABA (PYR/PYLs) and GA (GID1) receptors, explaining antagonistic role of these hormones in seed germination. While ABA promotes seed dormancy, GA peaks upon seed imbibition and initiates germination. In this work authors show that signaling by these hormones converge on RGL2, a DELLA repressor with a central role in inhibiting germination. Both ABA receptors (PYLs) and GA receptors (GID1s) GID1 directly interact with RGL2. PYLs stabilize RGL2 through the sequestration of DWA1, the substrate recognition of CUL4-DDB1 ubiquitin ligases ubiquitinating RGL2. GID1s overcome this stabilization by displacing PYLs, because higher binding affinity of the GID1-GA complex, which enables DWA1-mediated degradation of RGL2, in a synergistic fashion with the canonical GID1-SLY1 degradation pathway.

All 14 PYLs bound RGL2 in BiFC and in vitro pull-down studies. Moreover, ABA increased intensity of fluorescence interaction, GA consistently reducing complex formation in split-LUC assays. PYL4 overexpression reduced GA-induced RGL2 degradation, whereas RGL2 turnover was accelerated in the *pyr1 pyl124* quadruple mutant, in line with PYLs acting as RGL2 stabilizers. GID1-RGL2 interaction was in the presence of GA a lot stronger than that of PYR1/PYL4/11-RGL2 on ABA, showing that GA bound GID1 displaces PYLs-RGL2 interaction. Interestingly, presence of these hormones was essential in yeast three-hybrid competition studies, where GID1 did not interact with RGL2 in the absence of GA. Expression of PYR1/PYL4 inhibited formation of the GID1-RGL2 complexes, and ABA enhanced this suppression, consistent with a model whereby GA enhances GID1-RGL2 interaction and its competition ability, in contrast to ABA signaling that enhances PYLs inhibitory potential by inducing PYLs expression.

Both *pyr1 pyl124* quintuple mutants and *rgl2* knockouts are insensitive to ABA and PAC, while PYL4 overexpression leads to a hypersensitive response to both treatments. GID1 overexpression reduces response to both ABA and PAC, and this effect is partially suppressed by PYL4 oe. Moreover, the hypersensitive response of PYL4 oe is abolished in the *rgl2* background, showing that this effect depends on RGL2. From these results authors conclude that ABA and GA antagonistic effects on seed germination rely on competitive PYLs and GID1 interaction with RGL2. PYLs interaction stabilizes RGL2 by competing with GID1 binding, and ABA enhances this effect by promoting PYLs expression. GA in turn enhances GID1 interaction affinity and displaces PYLs from RGL2, allowing GA-induced proteasomal degradation of RGL2.

These findings nicely explain antagonistic effects of ABA and GAs on seed germination, while control of RGL2 stability provides a fully novel mechanism for this regulation. Nevertheless, role of DWA1 looks a lot less convincing. Authors demonstrate that DWA1 is a direct interactor of RGL2 in BiFC and co-IP assays. However, affinity of interaction is very minor as compared to DWA1-PYLs interaction (Fig 5B), with ABA further enhancing this interaction (Fig 5C). In *dwa1* null mutants, GA degradation of RGL2 is reduced (44% protein levels as compared to 24% in the wild-type), while it is accelerated in *dwa1* oe lines (48% as compared to 66% in the wild-type), although differences are not really striking. Authors show that DWA1 increases RGL2 ubiquitination, but this effect is again not dramatic. Notably, PYL4 inhibited DWA1-mediated RGL2-GFP ubiquitination (Fig. 6A), with RGL2 levels being similar in 35S:PYL4-GFP 35S:Flag-DWA1 as in 35S:PYL4-GFP lines, and higher than in the WT. PYL4 reduced the amount of MYC-DWA1 co-precipitated with RGL2-GFP (Fig. 6C) as well as fluorescence of BiFC interaction assays. More interestingly, PYL4 interfered with DWA1 and DDB1A/B interaction, and therefore DWA1 assembly into the CUL4 catalytic core, which authors concluded to play a role at enhancing RGL2 stability through DWA1 sequestration. Surprisingly, more MYC-DWA1 co-precipitated with RGL2-GFP under GA versus mock treatment, while this enhancement was nearly abolished in *gid1b/c* double mutants (Fig. 6G). However, it seems counterintuitive that DWA1 mediates RGL2 degradation in response to GA, when the F-box SLY is a very effective route for degradation in response to this hormone. Indeed, *dwa1* mutants are hypersensitive to ABA and PAC, while over-expression lines are resistant to ABA and PAC. PYL4 overexpression

restores ABA and PAC response of DWA1 *ox* lines, suggesting that this DWA1-dependent phenotype relies on PYL4. Did authors analyze PYL4 levels in DWA1*oe* and *dwa1* mutants? Indeed, DWA1 had been reported to act as a negative regulator in ABA signal transduction (Lee et al.; 2010), with ABI5 being more slowly degraded in *dwa1* and *dwa2* mutants. As such, it is well possible that PYL4 interference on DWA1-DDB1 interaction results in the stabilization of PYL4, whereas DWA1-PYL4 interaction also titrates out PYL4, decreasing its stabilizing effects on RGL2. This way, the *rgl2* mutation might suppress the PAC/ABA-hypersensitive phenotype of *dwa1*, via a similar effect as observed in Figure 3B for *rgl2* suppression of the PYL4 *oe* phenotype.

Overall, this is a very relevant study providing a novel molecular mechanism for ABA and GA antagonism on seed germination. The MS discussion covers however many aspects as a possible function of RGL2 in the cytosol, for which authors do not provide any evidence. It might be better to focus this section on the previous reports on DWA1 function as a negative regulator of ABA signaling, and the possible role of PYLs in regulating DWA1.

Minor points:

The three-hybrid experiment provided in Figure 2F is difficult to interpret. Some additional explanation would be required in the Figure legend. Shouldn't lower pictures corresponding to GA⁺ and ABA⁺, also be Met⁺?

Referee #1:

This manuscript elucidates a receptor competition paradigm in which RGL2 functions as a central hub for competitive receptor occupancy by the GA receptor GID1 and ABA receptors PYLs. This competition directly gates RGL2 degradation through the CUL4-DDB1 substrate receptor DWA1. The discoveries of hormone-tuned receptor competition and DWA1 as a receptor-regulated degradation conduit significantly advance our understanding of GA/ABA crosstalk beyond classical antagonism.

Major Comments:

Question 1. While PYLs-DWA1 direct physical interaction is demonstrated (Fig. 5), however, the molecular mechanism of the PYLs sequestration of DWA1 remains undefined. Do PYLs, RGL2, and DWA1 form a heterotrimeric complex? Do PYLs and RGL2 competitively bind to DWA1?

Suggestion: Perform competitive LCI and Co-IP assays to determine whether PYLs and RGL2 competitively interact with DWA1, and if *PYL4* overexpression rescues accelerated RGL2 degradation in *35S:Flag-DWA1* plants (linking sequestration to phenotype).

Response:

We appreciate the reviewer's insightful suggestion. To determine whether *PYL4* and RGL2 compete for DWA1 binding, we re-examined the interaction using two complementary approaches.

1. Competitive binding

MYC-DWA1 was immunoprecipitated from extracts co-expressing RGL2-GFP ± *PYL4*-mCherry (**revised Fig. 6C**). While both RGL2-GFP and *PYL4*-mCherry could be immunoprecipitated with DWA1, the presence of *PYL4*-mCherry reduced the amount of RGL2-GFP by ~80 % compared with the RGL2-only control, indicating that *PYL4* efficiently out-competes RGL2 for DWA1 association. This conclusion was corroborated by LCI (**Fig. 6D**): luciferase reconstitution between nLUC-RGL2 and cLUC-DWA1 was strongly attenuated when *PYL4*-mCherry, but not mCherry alone, was present. Together with the BiFC data, these results demonstrate competitive binding rather than formation of a stable heterotrimeric complex (**lines 212–217**).

2. Functional rescue

Cell-free degradation assays (**revised Fig. 6B, lines 210–212**) show that elevated DWA1 accelerates GA-triggered RGL2 turnover, whereas *PYL4* over-expression slows it. Importantly, *35S:Flag-DWA1 35S:PYL4* double-over-expressing extracts degrade RGL2 at a rate similar to *PYL4*-only extracts, indicating that *PYL4* can fully override the destabilizing effect of DWA1.

Collectively, our data support a model in which *PYL4* acts as a competitive sink for DWA1, thereby protecting RGL2 from ubiquitination and degradation.

Question 2. All 14 PYLs can interact with RGL2 and DWA1 (Fig. 1A, 5A), moreover, ABA treatment enhanced binding intensity, particularly for *PYL11* (Fig. 1C), but functional data (stabilization, competition) use only *PYL4*. The functional conservation among PYLs is unverified.

Suggestion: Include at least one extra PYL (e.g. *PYL11*) in degradation (Fig. 1D) and competition assays (Fig. 2C-F).

Response:

We thank the reviewer for raising this important point. To determine whether the RGL2-stabilising activity is conserved beyond PYL4, we crossed *35S:Flag-PYL11* (Zhao et al., 2020) to the *35S:MYC-RGL2* background and monitored RGL2 turnover in the resulting double-over-expression lines. As now shown in **revised Fig. 1D**, PYL11 retards GA-induced RGL2 degradation as PYL4, demonstrating that the protective function is not restricted to a single family member (**lines: 112-116**).

For the competition assays (**revised Fig. 2C–G**), we apologize for the previously confusing legend; the original panels already contained PYR1 and PYL4. Following your suggestion, we have now added PYL11. In both BiFC and yeast three-hybrid assays (also in pull down assay in **Fig. 2B**), PYL11 efficiently out-competes GID1 for RGL2 binding, yielding results similar to those obtained with PYR1 and PYL4 (**revised Fig. 2C–G, lines: 126-132**).

we apologize for the previously ambiguous figure legend; we have revised it for clarity. The original panels already included PYR1 and PYL4; following your advice we have now added PYL11. Consistent with the results obtained with PYR1 and PYL4, co-expression of PYL11 significantly suppresses the GID1–RGL2 interaction in both BiFC and Yeast three-hybrid assays

Collectively, these data confirm that at least three distinct PYL proteins (PYR1, PYL4 and PYL11) can stabilize RGL2 and disrupt GID1–RGL2 association, underscoring the functional conservation of this regulatory module across the PYL family.

References:

Zhao H, Nie K, Zhou H, Yan X, Zhan Q, Zheng Y, Song CP (2020) ABI5 modulates seed germination via feedback regulation of the expression of the PYR/PYL/RCAR ABA receptor genes. *New Phytol* 228:596–608.

Question 3. Quantitative interaction assays (LCI/BiFC) must be accompanied by expression validation of all fusion partners. Consequently, the experimental results regarding changes in interaction strength are scientifically invalid without expression controls (e.g. Figure 1C, D and Figure 2A, C, D, etc.).

Suggestion: Supplementary western blot results corresponding to the specific experiments must be provided, such as the nLUC-PYL and cLUC-RGL2 protein expression levels in Figure 1 C.

Response:

We thank the reviewer for emphasising this essential control. For every LCI panel we now provide corresponding immunoblots (anti-nLUC and anti-cLUC) to confirm equal expression of the fusion partners; the blots are presented in **Appendix Figs. S2, S4, S8 and S9** for **Fig. 1C, Fig. 2A, Fig. 5C, and Figs. 6D/F**, respectively.

For BiFC assays (**Figs. 2C, 2D, 6E and EV3**), we quantified transcript levels of YNE-, YCE- and mCherry-tagged constructs by RT–qPCR; no significant differences were detected (**Appendix Figs. S5 and S10**), verifying that changes in YFP fluorescence reflect genuine interaction differences rather than variable expression.

Likewise, in the RGL2 degradation time-courses (**Figs. 1D and 4C**), RT-qPCR demonstrates that *MYC-RGL2* transcript abundance remains constant throughout the assay (**Appendix Fig. S3**), validating that the observed changes in protein level are attributable to post-translational regulation.

These controls have been added to the revised manuscript; antibody details and primer sequences are provided in the Methods and Appendix Table S1.

Question 4. In Figure 4, there is a lack of genetic evidence demonstrating that DWA1 executes GA-dependent RGL2 ubiquitination.

Suggestion: Generate hybrid materials by crossing *35S:Flag:DWA1* transgenic lines with GA biosynthesis mutants to provide data on protein degradation rate and ubiquitination status of RGL2 in GA biosynthesis mutants.

Response:

Thank you for this insightful suggestion. To test whether DWA1-mediated RGL2 turnover is genetically dependent on endogenous GA biosynthesis, we monitored native RGL2 abundance in imbibed seeds of the GA-deficient mutant *gal-t* and *35S:Flag-DWA1 gal-t* using an anti-RGL2 antibody. Because native RGL2 levels in imbibed seeds are almost below the detection limit, we first stratified all seeds in 10 μ M PAC for 48 h at 4°C to synchronize and maximize RGL2 accumulation (Ariizumi and Steber, 2007; Piskurewicz et al., 2008); under these conditions the four genotypes contained equivalent amounts of RGL2 (time 0). PAC was then washed out and the seeds were allowed to germinate in water for 48 h in continuous light. Immunoblot analysis showed that RGL2 remained stable in *gal-t* seeds, whereas the *gal-t* background completely suppressed the accelerated turnover conferred by *35S:Flag-DWA1* (**Fig. 4F, lines:192-197**).

For ubiquitination analysis, after the same imbibition, seeds were germinated for 12 h in the presence of 50 μ M MG132. RGL2 was immunoprecipitated with anti-RGL2 antibody and probed with anti-ubiquitin antibody. Our results showed that ubiquitin-conjugated RGL2 was undetectable in *gal-t* and that DWA1-enhanced ubiquitination was abolished in *35S:Flag-DWA1 gal-t* (**Fig. 4G, lines:192-197**).

Thus, DWA1 cannot destabilize RGL2 when GA biosynthesis is blocked, providing genetic evidence that DWA1 executes GA-dependent ubiquitination and degradation of RGL2.

References:

- Ariizumi T, Steber CM (2007) Seed germination of GA-insensitive *sleepy1* mutants does not require RGL2 protein disappearance in Arabidopsis. *Plant Cell* 19:791–804.
- Piskurewicz U, Jikumaru Y, Kinoshita N, Nambara E, Kamiya Y, Lopez-Molina L (2008) The gibberellic acid signaling repressor RGL2 inhibits Arabidopsis seed germination by stimulating abscisic acid synthesis and ABI5 activity. *Plant Cell* 20:2729–2745.

Question 5. In Figure 4E, I notice the absence of detectable change of RGL2 ubiquitination levels in both WT and *35S:Flag:DWA1* seedlings without exogenous GA treatment, how does this occur in the presence of endogenous GA?

Suggestion: WT and *35S:Flag:DWA1* seedlings could undergo MG132+PAC \pm GA co-treatments to eliminate confounding effects of endogenous GA on DWA1-mediated RGL2 ubiquitination levels.

Response:

We thank the reviewer for this perceptive comment. To uncouple endogenous GA from the assay, we first pre-treated seedlings with PAC to abolish endogenous GA biosynthesis; the resulting ubiquitinated-RGL2 signal was too faint to discriminate between WT and *35S:Flag-DWA1*

samples, likely because GA levels were driven below the threshold required for efficient DELLA ubiquitination. We therefore adopted a protocol in which seedlings were pre-incubated with 50 μ M MG132 for 6 h to accumulate ubiquitinated forms without perturbing internal GA levels. Under these conditions we detected a moderate but consistent increase in high-molecular-weight RGL2 species in *35S:Flag-DWA1* extracts compared with WT (**revised Fig. 4E, -GA lanes**). When 10 μ M GA + 50 μ M MG132 was subsequently supplied for 3 h, ubiquitination was strongly enhanced in both genotypes, yet the *35S:Flag-DWA1* samples again exhibited dramatically more ubiquitinated RGL2 (**revised Fig. 4E, +GA lanes**). Thus, DWA1 enhances RGL2 ubiquitination in proportion to available GA, whether endogenous or exogenously supplied.

Question 6. In Figure 6B, I find substantial variation in the initial MBP-RGL2 protein levels among seedling protein extracts, Furthermore, all MBP-RGL2 Western blot results should be displayed on a single membrane to reliably demonstrate degradation rate differences.

Suggestion: Load the protein samples containing MBP-RGL2 fusion protein on a single membrane for comparative analysis while adding more time points (0, 20, 40, 60 min).

Response:

We thank the reviewer for this critical suggestion. We repeated the cell-free assay with all MBP-RGL2 samples loaded on a single membrane and extended the time course to 0, 20, 40 and 60 min (**revised Fig. 6B** and **new Fig. 6I**). Equal loading was verified by Ponceau-S staining (lower panels). MBP-RGL2 declined significantly faster in *35S:Flag-DWA1* extracts than in WT, whereas *35S:PYL4-GFP* extracts retained more substrate at every time point. The double-over-expressing line behaved similarly to the PYL4-only line, confirming that PYL4 overrides DWA1-accelerated degradation.

Question 7. The manuscript aims to elucidate the competitive interplay between RGL2 and GID1 versus PYL, revealing a mechanism distinct from the classical GA/ABA antagonism in seed germination regulation. While PYLs directly stabilize RGL2 is demonstrated (Figure 1), how GID1 directly destabilize RGL2 remains unclear (mediated by GID1-DWA module, distinct from the canonical GID1-SLY1 pathway).

Suggestion: Perform assays to demonstrate GID1 promotion of DWA1-mediated ubiquitination (like Figure 6A) and compare MBP-RGL2 degradation rates among extracts from WT, *Flag-DWA1*, *gid1a gid1b gid1c* and *gid1a gid1b gid1c Flag-DWA1* seeding (like Figure 6).

Response:

We thank the reviewer for this insightful suggestion. In vitro ubiquitination assays show that addition of recombinant GID1a or GID1b (+GA) strongly enhances DWA1-dependent poly-ubiquitination of RGL2 (**updated Fig. 6H**), demonstrating that GID1 is an active component of the DWA1 module.

Genetically, repeated attempts to introduce *35S:Flag-DWA1* into the *gid1a/b/c* triple mutant were unsuccessful owing to severe growth defects; we therefore analysed the available *gid1b/c* double mutant. Cell-free degradation assays (0–60 min, single membrane) revealed that GA-triggered turnover of MBP-RGL2 is already markedly slowed in *gid1b/c* extracts and that *Flag-DWA1* no longer accelerates degradation in this background (**Fig. 6I, lines 225–228**).

Thus, GID1 activity is indispensable for DWA1-mediated, GA-dependent degradation of RGL2,

supporting a model in which GID1 promotes substrate recruitment to the DWA1 ubiquitination machinery independently of the canonical SLY1 pathway.

Question 8. Following the previous question, how to distinguish DWA1-mediated RGL2 degradation with canonical GA-GID1SLY1 pathways?

Suggestion: Perform assays to demonstrate if GA-DWA1-mediated RGL2 degradation exists in *sly1* mutant.

Response:

We thank the reviewer for this important point. The relationship between DWA1 and SLY1 is also a subject of great interest to us.

To test whether DWA1 acts independently of SLY1, we compared native RGL2 levels in *dwa1*, *sly1* and *35S:Flag-DWA1 sly1* germinating seeds. Immunoblots showed that *sly1* retained abundant RGL2, whereas *dwa1* contained only trace amounts, confirming that SLY1 is the principal germination-triggered E3 (**Fig. EV4A**). However, DWA1 over-expression modestly accelerated RGL2 turnover even in the *sly1* background (**Fig. EV4B**), and the two proteins interact in vivo without altering SLY1 abundance (**Fig. EV4 C–E**).

These observations, together with DWA1's WD40 propeller structure, suggest that DWA1 does not replace SLY1 but acts as a scaffold that facilitates SLY1-mediated recognition or ubiquitin transfer when environmental cues demand fine-tuned control of DELLA abundance. Thus, our data position DWA1 as a context-dependent modulator that operates alongside, but distinct from, SLY1. SLY1 provides the constitutive “on/off” switch for germination, while DWA1 serves as an adjustable rheostat that integrates hormone and stress signals to set the speed of RGL2 removal. We have incorporated these conclusions into the Discussion (**Lines: 275–291**).

Minor Comments:

Question 1. The manuscript contains several labeling inaccuracies that require correction. Please verify whether nLUC-PYLs+cLUC-DWA1+ABA and nLUC-PYLs+cLUC-DWA1+GA should be nLUC-PYLs+cLUC-RGL2+ABA and nLUC-PYLs+cLUC-RGL2+GA due to a labeling error in Figure 1C.

Response:

We apologize for this labeling error. The correct combinations are nLUC-PYLs + cLUC-RGL2 + ABA and nLUC-PYLs + cLUC-RGL2 + GA. We have amended the labels in Figure 1C and revised the labels throughout the manuscript.

Question 2. To compare the differences in MYC-RGL2 protein stability in different seedlings in Figure 1D and Figure 4C, it is necessary to provide qPCR data demonstrating consistent transcriptional levels of MYC-RGL2.

Response:

Thank you for this important suggestion. RT-qPCR of every genotype and time point used in Figures 1D and 4C shows that MYC-RGL2 transcript levels remain constant (**Appendix Fig. S3**, $P > 0.05$, one-way ANOVA), confirming that the observed protein differences reflect post-translational regulation.

Question 3. In Figure 2B, to demonstrate the difference in binding capacity between RGL2 and GID1 versus PYL, a more robust approach is to use either MST (MicroScale Thermophoresis) or ITC (Isothermal Titration Calorimetry) experiments to quantify the differences in their affinity constants.

Response:

We thank the reviewer for this valuable suggestion. To quantify the binding difference we performed MST by titrating recombinant GST-GID1a/b or GST-PYLS against sfGFP-RGL2 (intrinsic fluorescence) in the presence of saturating GA (100 μ M) or ABA (100 μ M), respectively (**Fig. EV1, lines 126-132**).

Under these conditions GID1a/b gave apparent Kd values of $3.9 \pm 1.5 \mu\text{M}$ and $4.6 \pm 0.7 \mu\text{M}$, whereas all PYLS were markedly weaker (PYR1: $44 \pm 26 \mu\text{M}$; PYL4: $23 \pm 13 \mu\text{M}$; PYL11: $17 \pm 9 \mu\text{M}$). Thus, MST qualitatively confirms that GID1 binds RGL2 more tightly than any PYL, providing a biochemical basis for the competitive displacement observed in vivo.

We fully recognize that these absolute affinities are weaker than the nanomolar Kd values reported by QCM or SPR (Suzuki et al., 2006; Hirano et al., 2010). This discrepancy is attributable to:

1. High-salt buffer (250 mM NaCl) required to keep sfGFP-RGL2 monomeric, the elevated ionic strength weakens electrostatic contacts and limits GST-GID1/PYL solubility, so visible precipitation occurred above $\sim 100 \mu\text{M}$ ligand.
2. The N-terminal GST tag, which may introduce steric/electrostatic perturbations.

Consequently, binding isotherms failed to reach full saturation, yielding upper-limit Kd estimates with enlarged confidence intervals. We therefore present the MST data only as supplementary information (**Fig. EV1**) and emphasize the qualitative ranking (GID1 > PYL).

To further corroborate this hierarchy we have added a GST-alone negative control and PYL11 (the strongest PYL interactor) to **Fig. 2B**; even PYL11 gives a pull-down signal only $\sim 25\%$ of that obtained with GID1, underscoring the robustness of the difference. Should the reviewer prefer, we will gladly remove the MST panel, but we hope the combined in-vitro and pull-down evidence now satisfactorily supports the relative affinity conclusion.

References:

- Hirano K, Asano K, Tsuji H, Kawamura M, Mori H, Kitano H, Ueguchi-Tanaka M, Matsuoka M. (2010) Characterization of the molecular mechanism underlying gibberellin perception complex formation in rice. *Plant Cell* 22:2680-2096.
- Suzuki H, Park SH, Okubo K, Kitamura J, Ueguchi-Tanaka M, Iuchi S, Katoh E, Kobayashi M, Yamaguchi I, Matsuoka M, et al (2009) Differential expression and affinities of Arabidopsis gibberellin receptors can explain variation in phenotypes of multiple knock-out mutants. *Plant J* 60:48–55.

Question 4. *pyr1 pyl124* quadruple mutants should be *pyr1 pyl1 pyl2 pyl4* (list all genes) according to Park et al., 2009.

Response:

Thank you for this correction. We have changed the genotype designation to *pyr1 pyl1 pyl2 pyl4* throughout the manuscript, figures and figure legends, exactly as reported in Park et al., 2009.

Referee #2:

Nie et al report in this manuscript a competitive mechanism for RGL2 stabilization/ proteasomal degradation by the ABA (PYR/PYLs) and GA (GID1) receptors, explaining antagonistic role of these hormones in seed germination. While ABA promotes seed dormancy, GA peaks upon seed imbibition and initiates germination. In this work authors show that signaling by these hormones converge on RGL2, a DELLA repressor with a central role in inhibiting germination. Both ABA receptors (PYLs) and GA receptors (GID1s) GID1 directly interact with RGL2. PYLs stabilize RGL2 through the sequestration of DWA1, the substrate recognition of CUL4-DDB1 ubiquitin ligases ubiquitinating RGL2. GID1s overcome this stabilization by displacing PYLs, because higher binding affinity of the GID1-GA complex, which enables DWA1-mediated degradation of RGL2, in a synergistic fashion with the canonical GID1-SLY1 degradation pathway.

All 14 PYLs bound RGL2 in BiFC and in vitro pull-down studies. Moreover, ABA increased intensity of fluorescence interaction, GA consistently reducing complex formation in split-LUC assays. PYL4 overexpression reduced GA-induced RGL2 degradation, whereas RGL2 turnover was accelerated in the *pyr1 pyl124* quadruple mutant, in line with PYLs acting as RGL2 stabilizers. GID1-RGL2 interaction was in the presence of GA a lot stronger than that of PYR1/PYL4/11-RGL2 on ABA, showing that GA bound GID1 displaces PYLs-RGL2 interaction. Interestingly, presence of these hormones was essential in yeast three-hybrid competition studies, where GID1 did not interact with RGL2 in the absence of GA. Expression of PYR1/PYL4 inhibited formation of the GID1-RGL2 complexes, and ABA enhanced this suppression, consistent with a model whereby GA enhances GID1-RGL2 interaction and its competition ability, in contrast to ABA signaling that enhances PYLs inhibitory potential by inducing PYLs expression.

Both *pyr1 pyl124* quintuple mutants and *rgl2* knockouts are insensitive to ABA and PAC, while *PYL4* overexpression leads to a hypersensitive response to both treatments. *GID1* overexpression reduces response to both ABA and PAC, and this effect is partially suppressed by *PYL4* oe. Moreover, the hypersensitive response of *PYL4* oe is abolished in the *rgl2* background, showing that this effect depends on RGL2. From these results authors conclude that ABA and GA antagonistic effects on seed germination rely on competitive PYLs and GID1 interaction with RGL2. PYLs interaction stabilizes RGL2 by competing with GID1 binding, and ABA enhances this effect by promoting PYLs expression. GA in turn enhances GID1 interaction affinity and displaces PYLs from RGL2, allowing GA-induced proteasomal degradation of RGL2.

These findings nicely explain antagonistic effects of ABA and GAs on seed germination, while control of RGL2 stability provides a fully novel mechanism for this regulation. Nevertheless, role of DWA1 looks a lot less convincing. Authors demonstrate that DWA1 is a direct interactor of RGL2 in BiFC and co-IP assays.

However, affinity of interaction is very minor as compared to DWA1-PYLs interaction (Fig 5B), with ABA further enhancing this interaction (Fig 5C). In *dwa1* null mutants, GA degradation of RGL2 is reduced (44% protein levels as compared to 24% in the wild-type), while it is accelerated in *DWA1* oe lines (48% as compared to 66% in the wild-type), although differences are not really striking. Authors show that *DWA1* increases RGL2 ubiquitination, but this effect is again not dramatic.

Notably, *PYL4* inhibited *DWA1*-mediated RGL2-GFP ubiquitination (Fig. 6A), with RGL2

levels being similar in 35S:PYL4-GFP 35S:Flag-DWA1 as in 35S:PYL4-GFP lines, and higher than in the WT. PYL4 reduced the amount of MYC-DWA1 co-precipitated with RGL2-GFP (Fig. 6C) as well as fluorescence of BiFC interaction assays. More interestingly, PYL4 interfered with DWA1 and DDB1A/B interaction, and therefore DWA1 assembly into the CUL4 catalytic core, which authors concluded to play a role at enhancing RGL2 stability through DWA1 sequestration.

Surprisingly, more MYC-DWA1 co-precipitated with RGL2-GFP under GA versus mock treatment, while this enhancement was nearly abolished in *gid1b/c* double mutants (Fig. 6G).

However, it seems counterintuitive that DWA1 mediates RGL2 degradation in response to GA, when the F-box SLY is a very effective route for degradation in response to this hormone. Indeed, *dwa1* mutants are hypersensitive to ABA and PAC, while over-expression lines are resistant to ABA and PAC. PYL4 overexpression restores ABA and PAC response of DWA1 oe lines, suggesting that this DWA1-dependent phenotype relies on PYL4.

Did authors analyze PYL4 levels in *DWA1oe* and *dwa1* mutants? Indeed, DWA1 had been reported to act as a negative regulator in ABA signal transduction (Lee et al.; 2010), with ABI5 being more slowly degraded in *dwa1* and *dwa2* mutants. As such, it is well possible that PYL4 interference on DWA1-DDB1 interaction results in the stabilization of ABI5, whereas DWA1-PYL4 interaction also titrates out PYL4, decreasing its stabilizing effects on RGL2. This way, the *rgl2* mutation might suppress the PAC/ABA-hypersensitive phenotype of *dwa1*, via a similar effect as observed in Figure 3B for *rgl2* suppression of the PYL4 oe phenotype.

Overall, this is a very relevant study providing a novel molecular mechanism for ABA and GA antagonism on seed germination. The MS discussion covers however many aspects as a possible function of RGL2 in the cytosol, for which authors do not provide any evidence. It might be better to focus this section on the previous reports on DWA1 function as a negative regulator of ABA signaling, and the possible role of PYLs in regulating DWA1.

We sincerely appreciate the reviewer's endorsement of our proposed mechanism. To address the remaining concerns, we first reproduce the relevant original comments, concisely summarize each question, and then provide new experimental evidence or reasoned interpretations. We hope these additional results and clarifications will fully alleviate any doubts about our work.

"However, affinity of interaction is very minor as compared to DWA1-PYLs interaction (Fig 5B), with ABA further enhancing this interaction (Fig 5C). In *dwa1* null mutants, GA degradation of RGL2 degradation is reduced (44% protein levels as compared to 24% in the wild-type), while it is accelerated in DWA1 oe lines (48% as compared to 66% in the wild-type), although differences are not really striking. Authors show that DWA1 increases RGL2 ubiquitination, but this effect is again not dramatic."

"However, it seems counterintuitive that DWA1 mediates RGL2 degradation in response to GA, when the F-box SLY is a very effective route for degradation in response to this hormone."

Question 1. The position of DWA1 in GA-induced RGL2 degradation. The interaction between DWA1 and RGL2 is minor, and the DWA1-mediated ubiquitination and degradation of RGL2 is not dramatic. Moreover, the relationship with the known GID1-SLY1 pathway is unclear.

Response:

We fully agree with the reviewer that the DWA1–RGL2 interaction is weak and that the resulting degradation is modest. Indeed, we repeatedly obtain only a faint signal, consistent with the transient nature of most E3–substrate pairs. Nevertheless, this interaction was consistently detected in four independent assays (BiFC, pull-down, LCI and Co-IP; **Fig. 4 A–C** and **Fig. 6C**), and we observe that GA strongly enhances the association. Moreover, when GID1a/b is supplied together with GA, DWA1-mediated RGL2 ubiquitination is markedly amplified (**new Fig. 6H**), indicating that GA-bound GID1 is an integral component of the recognition complex.

To address the degradation concern more directly, we have now added cell-free assays that demonstrate a clear, albeit moderate, enhancement of GA-induced RGL2 degradation by DWA1 (**new Fig. 6, B and I**). We therefore view DWA1 as a context-dependent modulator that acts alongside—rather than replaces—the canonical SLY1 pathway. Consistent with this interpretation, immunoblots show that germinating *dwa1* seeds retain only trace RGL2, whereas *sly1* seeds contain abundant RGL2 (**Fig. EV4A**), confirming that SLY1 is the principal germination-triggered E3. Loss of DWA1 merely delays RGL2 disappearance, whereas loss of SLY1 almost blocks it. Nevertheless, DWA1 over-expression can still modestly accelerate RGL2 turnover in the *sly1* background (**Fig. EV4B**), and the two E3 components interact in vivo without altering SLY1's abundance (**Fig. EV4 C–E**).

Taken together, these data lead us to propose that SLY1 provides the constitutive “on/off” switch for germination, while DWA1 functions as an adjustable rheostat that integrates hormone and stress signals to fine-tune the rate of RGL2 removal. We have incorporated this discussion into the revised manuscript (**lines 275–291**).

“Surprisingly, more MYC-DWA1 co-precipitated with RGL2-GFP under GA versus mock treatment, while this enhancement was nearly abolished in *gid1b/c* double mutants (Fig. 6G).”

Question 2. The role of GA receptor GID1.

Response:

We thank the reviewer for highlighting this important point. To clarify the role of GID1, we have now provided direct evidence that the GA receptor is indispensable for DWA1-mediated RGL2 turnover. In newly added in-vitro ubiquitination assays, the inclusion of recombinant GID1a or GID1b together with GA strongly stimulated DWA1-dependent ubiquitination of RGL2 (**Fig. 6H**). Consistently, the accelerated degradation conferred by *35S:Flag-DWA1* was almost completely abolished in the *gid1b/c* double-mutant background (**Fig. 6I**), demonstrating that GID1 activity is required for DWA1 to exert its effect on RGL2 stability. We believe these data reinforce the notion that GID1 acts as an essential component of the DWA1–RGL2 recognition complex.

“Did authors analyze PYL4 levels in *DWA1oe* and *dwa1* mutants? Indeed, DWA1 had been reported to act as a negative regulator in ABA signal transduction (Lee et al.; 2010), with ABI5 being more slowly degraded in *dwa1* and *dwa2* mutants. As such, it is well possible that PYL4 interference on DWA1-DDB1 interaction results in the stabilization of ABI5, whereas DWA1-PYL4 interaction also titrates out PYL4, decreasing its stabilizing effects on RGL2. This

way, the *rgl2* mutation might suppress the PAC/ABA-hypersensitive phenotype of *dwa1*, via a similar effect as observed in Figure 3B for *rgl2* suppression of the *PYL4* oe phenotype.”

Question 3. Whether DWA1 can reduce the protein level or titrate out *PYL4* by interaction, thus indirectly weakening the stabilizing effect of *PYL4* on *RGL2*?

Response:

We appreciate the reviewer’s insightful suggestion that DWA1 might lower free *PYL4* levels and thereby indirectly reduce *RGL2* stability. To test this possibility we first compared *PYL4* protein abundance in WT, *dwa1* null and *35S:Flag-DWA1* lines. We observed no consistent change in *PYL4* levels (showed below), indicating that the interaction does not lead to *PYL4* degradation or “titration-out”.

Figure for referee with unpublished data and its description has been removed upon request by the authors.

We believe this stability is consistent with our working model: *PYL4* binds DWA1 at the expense of DWA1–DDB1 association, so any decrease in “free” *PYL4* is automatically mirrored by a decrease in functional DWA1. Consequently, the *PYL4*:DWA1 ratio—and thus the protection of *RGL2*—remains essentially unchanged.

Functional data support this interpretation. In cell-free degradation assays, *DWA1-OX* accelerates GA-induced *RGL2* turnover, whereas *PYL4-OX* completely blocks this acceleration; the double-over-expression line behaves identically to *PYL4-OX* alone (**Fig. 6B**), placing *PYL4* genetically upstream of (and epistatic to) DWA1.

Similarly, at the whole-seed level, *dwa1 rgl2* double mutants germinate like *rgl2* single mutants under ABA or PAC (**Fig. 7, lane 7**), and *35S:PYL4-GFP 35S:Flag-DWA1* double-over-expression lines respond to ABA/PAC indistinguishably from *PYL4-OX* alone (**Fig. 7, lane 9**). Taken together, these results argue that DWA1 does not appreciably reduce *PYL4* protein levels; rather, *PYL4* governs the availability of DWA1, thereby setting the pace of *RGL2* degradation.

Question 4. The MS discussion covers however many aspects as a possible function of *RGL2* in the cytosol, for which authors do not provide any evidence. It might be better to focus this section on the previous reports on DWA1 function as a negative regulator of ABA signaling, and the possible role of *PYLs* in regulating DWA1.

Response:

We thank the reviewer for this helpful comment. We fully agree that the original discussion

over-extended into speculative cytosolic functions of RGL2 for which we currently lack data. Accordingly, we have deleted that section and inserted a focused paragraph (**lines 292-305**) that:

1. Summarizes the previously reported role of DWA1 as a negative regulator of ABA signaling through ABI5 destabilization (Lee et al., 2010);
2. Integrates our new finding that PYLs competitively sequester DWA1, thereby modulating its availability for ABI5 and RGL2 turnover;
3. Emphasizes how this dual control links GA and ABA pathways during seed development.

We believe this revision keeps the discussion firmly grounded in evidence while highlighting the mechanistic unity of DWA1-PYL regulation across both previously published and present datasets.

Minor points:

The three-hybrid experiment provided in Figure 2F is difficult to interpret. Some additional explanation would be required in the Figure legend. Shouldn't lower pictures corresponding to GA⁺ and ABA⁺, also be Met⁺?

Response:

We apologise for the confusing layout of the original Fig. 2F. Following the reviewer's suggestion we have completely reorganised the panel (**new Fig. 2F, G**), inserted a vector schematic (**Appendix Fig. S6**) and rewritten the legend for clarity. A step-by-step description is now provided in the figure legend; the key points are summarised below.

1. Construct design

- BD-PYL-GID1: *GID1* expression is under the *MET25* promoter → induced only in Met-free medium (Met-).
- BD-GID1-PYL: *PYL* expression is under the *MET25* promoter → induced only in Met-medium.

Standard SD medium contains methionine (Met⁺); Met-free medium is labelled Met-.

2. Panel layout (new Fig. 2F, G)

Left (Met⁺): basal interaction controls.

2F: BD-PYL + AD-RGL2 (PYL-RGL2 interaction).

2G: BD-GID1 + AD-RGL2 (GID1-RGL2 interaction).

Middle (Met-/no hormone): competitor protein induced.

2F: BD-PYL-GID1 (Met-) + AD-RGL2 → GID1 is present; without GA, GID1 does not bind RGL2, so the PYL-RGL2 interaction is unchanged versus Met⁺.

2G: BD-GID1-PYL (Met-) + AD-RGL2 → PYL is present; PYL binds RGL2 and therefore weakens the GID1-RGL2 signal.

Right (Met-/hormone⁺):

2F (GA⁺): GA triggers GID1-RGL2 association, which out-competes PYL-RGL2 and reduces the PYL-RGL2 signal.

2G (ABA⁺): ABA enhances PYL-RGL2 affinity; the already-induced PYL more effectively excludes GID1, further reducing the GID1-RGL2 signal.

All negative controls (empty vectors) have been moved to **Source Data Fig. 2F, G** to keep the main figure concise while maintaining full transparency.

Dear Prof. Zheng,

We have now received re-review reports both referees, which I have included below. As you will see, you have addressed their concerns satisfactorily. Before I can finally accept the manuscript, there are some remaining editorial points which need to be addressed. In this regard would you please:

- remove EV figures from the main manuscript,
- ensure an institutional email address is used for corresponding authors, note that employment in a biotech company should be stated in the "Disclosure and competing interests statement",
- remove the AC/CrediT section from the text,
- include a callout in the text for Appendix Table S1,
- upload EV figures as individual, high-resolution Figure files with legends remaining in the manuscript below the main figure legends,
- include page numbers for the listed items in the table of contents on the appendix title page,
- be reminded that there is the opportunity to attach any lab protocols to the manuscript that you think might be helpful to the community,
- removed the Reagents and Tools table from the main manuscript and upload it as an individual file using the template from our guide to authors,
- save Source data in a scheme of one figure per folder and then upload as .zip files. E.g. all the Source data files for figure 1 need to be saved in a single folder and this needs to be zipped and then uploaded as "SD figure 1.zip" file. For EV and/or appendix figures, ZIP together all source data,
- provide Source Data for Appendix Fig S1,
- provide exact p values in the legends of figures 1C, D; 2A, E; 3B, 4C, 5C, 6D, F; 7B,
- define the error bars in the legend of figure 4C, and
- correct the section order as follows: Title page - Abstract - Keywords - Introduction - Results - Discussion - Methods - Data Availability - Acknowledgements - Disclosure and Competing Interests Statement - References - Figure Legends - Table(s) - Expanded View Figure Legends.

We include a synopsis of the paper (see <http://emboj.embopress.org/>). Please provide me with a general summary image, a two-sentence summary statement and 3-5 bullet points that capture the key findings of the paper.

I am looking forward to receiving your revised manuscript.

EMBO Press is an editorially independent publishing platform for the development of EMBO scientific publications.

Best wishes,

William Teale

William Teale, PhD
Editor
The EMBO Journal
w.teale@embojournal.org

Read our guidance for manuscript revisions and related editorial policies: <https://link.springer.com/journal/44318/submission-guidelines#cms-Revised-submissions>

<https://media.springernature.com/original/springer-cms/rest/v1/content/27825798/data/v1>

- a point-by-point response to the referees' comments, with a detailed description of the changes made (as a word file).
- a word file of the manuscript text.
- individual production quality figure files (one file per figure)

- a complete author checklist
- Expanded View files (replacing Supplementary Information)
- a Reagents and Tools Table as part of the Methods section

Please remember: Digital image enhancement is acceptable practice, as long as it accurately represents the original data and conforms to community standards. If a figure has been subjected to significant electronic manipulation, this must be noted in the figure legend or in the 'Methods' section. The editors reserve the right to request original versions of figures and the original images that were used to assemble the figure.

We realize that it is difficult to revise to a specific deadline. In the interest of protecting the conceptual advance provided by the work, we recommend a revision within 3 months (4th Mar 2026). Please discuss the revision progress ahead of this time with the editor if you require more time to complete the revisions.

Referee #1:

This revised manuscript has addressed mostly my previous comments and concerns. I suggest its acceptance by the journal.

Referee #2:

Nie et al. identify in this manuscript the ABA receptor PYLs as direct binding partners of RGL2 and describe a receptor-competition paradigm whereby PYLs stabilize RGL2 through physical interaction and sequestration of DWA1, the substrate recognition component of the CUL4-DDB1 E3 complex regulating RGL2 ubiquitination. The GA receptors GID1 counteract this stabilization by binding RGL2 with higher affinity, thereby displacing PYLs and promoting seed germination through enhanced DWA1-mediated RGL2 degradation.

In the revised manuscript, authors have added protein expression controls for all BiFC and split-LUC assays and provide evidence that PYL4 outcompetes RGL2 for association with DWA1. They further demonstrate that DWA1 accelerates GA-triggered RGL2 turnover, with additional experiments supporting that this regulatory mechanism extends to all PYLs and that DWA1 mediates GA-dependent RGL2 ubiquitination. Importantly, GID1 activity is essential for DWA1-accelerated RGL2 degradation, whereas they show that DWA1 functions independently of SLY1 and its effect on RGL2 destabilization are not an indirect consequence of reduced PYL abundance.

Taken together, the authors have fully addressed the reviewers' concerns, and the revised manuscript now provides a compelling mechanistic framework that supports acceptance for publication.

Dear Prof. Zheng,

We have now received re-review reports both referees, which I have included below. As you will see, you have addressed their concerns satisfactorily. Before I can finally accept the manuscript, there are some remaining editorial points which need to be addressed. In this regard would you please:

1. Remove EV figures from the main manuscript. Done.

2. Ensure an institutional email address is used for corresponding authors. Done.

3. Note that employment in a biotech company should be stated in the "Disclosure and competing interests statement".

We have verified that none of the authors are employed by a biotech company.

4. Remove the AC/CrediT section from the text. Done.

5. Include a callout in the text for Appendix Table S1.

We have added a citation to Appendix Table S1 in the "BiFC and LCI assays" subsection of the Methods.

6 Upload EV figures as individual, high-resolution Figure files with legends remaining in the manuscript below the main figure legends.

Done.

7 Include page numbers for the listed items in the table of contents on the appendix title page.

Done.

8 Be reminded that there is the opportunity to attach any lab protocols to the manuscript that you think might be helpful to the community.

All necessary protocols have been included in the Methods section.

9 Removed the Reagents and Tools table from the main manuscript and upload it as an individual file using the template from our guide to authors.

Done.

10 Save Source data in a scheme of one figure per folder and then upload as .zip files. E.g. all the Source data files for figure 1 need to be saved in a single folder and this needs to be zipped and then uploaded as "SD figure 1.zip" file. For EV and/or appendix figures, ZIP together all source data.

Done.

11 Provide Source Data for Appendix Fig S1. Done.

12 Provide exact p values in the legends of figures 1C, D; 2A, E; 3B, 4C, 5C, 6D, F; 7B.

Upon reviewing recent publications in “The EMBO Journal”, we observed that exact p values are typically presented within the figures themselves rather than in the legends. Following this common practice, we have added exact p values directly to the figures 1C, D; 2E; 4C, 5C, 6D, F.

For figures 2A, 3B, and 7B, after careful consideration we have retained the original letter-based annotation system. These figures involve extensive multiple comparisons (Figure 2A: 6 groups generating 15 p values; Figure 3B: 7 groups under 3 conditions generating 61 p values; Figure 7B: 9 groups under 3 conditions generating 108 p values), making it impractical to clearly display all exact p values within the figure or legend due to space constraints. All exact p values are fully documented in the corresponding Source Data files. If required, we can modify these figures to show p values for comparisons with WT only, though this would omit the comprehensive pairwise comparison information currently conveyed by the letter system.

13 Define the error bars in the legend of figure 4C. Done.

14 Correct the section order as follows: Title page - Abstract - Keywords - Introduction - Results - Discussion - Methods - Data Availability - Acknowledgements - Disclosure and Competing Interests Statement - References - Figure Legends - Table(s) - Expanded View Figure Legends.

Done.

15 We include a synopsis of the paper (see <http://emboj.embopress.org/>). Please provide me with a general summary image, a two-sentence summary statement and 3-5 bullet points that capture the key findings of the paper.

Done.

I am looking forward to receiving your revised manuscript.

EMBO Press is an editorially independent publishing platform for the development of EMBO scientific publications.

Best wishes,

William Teale

William Teale, PhD

Editor

The EMBO Journal

w.teale@embojournal.org

Dear Prof. Zheng,

I am pleased to inform you that your manuscript has been accepted for publication in the EMBO Journal.

Congratulations to you and your team!

You may qualify for financial assistance for your publication charges - either via a Springer Nature fully open access agreement or an EMBO initiative. Check your eligibility: <https://link.springer.com/journal/44318/how-to-publish-with-us>

Yours sincerely,

William Teale

William Teale, PhD
Editor
The EMBO Journal
w.teale@embojournal.org

Please note that it is The EMBO Journal policy for the transcript of the editorial process (containing referee reports and your response letters) to be published as an online supplement to each paper. If you should prefer removal of any referee-only figures included in the point-by-point response(s), e.g. because they may still be used for future publication or because they have been reproduced from published work by others, please do let us know immediately via response email.

More information is available here: <https://link.springer.com/partners/embo-press/editorial-policies#Peer%20review>